# Biochemical and Functional Characterization of the Three Zebrafish Transglutaminases 2

**DOI:** 10.3390/ijms241512041

**Published:** 2023-07-27

**Authors:** Manuel Lisetto, Mariagiulia Fattorini, Andrea Lanza, Marco Gerdol, Martin Griffin, Zhuo Wang, Fortunato Ferrara, Daniele Sblattero

**Affiliations:** 1Department of Life Sciences, University of Trieste, 34127 Trieste, Italy; manuel.lisetto@phd.units.it (M.L.); mariagiulia.fattorini@phd.units.it (M.F.); andrea.lanza2@phd.units.it (A.L.); mgerdol@units.it (M.G.); 2College of Health and Life Sciences, Aston University, Aston Triangle, Birmingham B4 7ET, UK; m.griffin@aston.ac.uk (M.G.); z.wang10@aston.ac.uk (Z.W.); 3Specifica a Q2 Solutions Company, Santa Fe, NM 87501, USA; fortunato.ferrara@q2labsolutions.com

**Keywords:** transglutaminase, zebrafish, cell adhesion, apoptosis, enzyme

## Abstract

Transglutaminase 2 (TG2) is a multifunctional protein widely distributed in various tissues and involved in many physiological and pathological processes. However, its actual role in biological processes is often controversial as TG2 shows different effects in these processes depending on its localization, cell type, or experimental conditions. We characterized the enzymatic and functional properties of TG2 proteins expressed in Danio rerio (zebrafish) to provide the basis for using this established animal model as a reliable tool to characterize TG2 functions in vivo. We confirmed the existence of three genes orthologous to human TG2 (zTGs2) in the zebrafish genome and their expression and function during embryonic development. We produced and purified the zTGs2s as recombinant proteins and showed that, like the human enzyme, zTGs2 catalyzes a Ca^2+^ dependent transamidation reaction that can be inhibited with TG2-specific inhibitors. In a cell model of human fibroblasts, we also demonstrated that zTGs2 can mediate RGD-independent cell adhesion in the extracellular environment. Finally, we transfected and selected zTGs2-overexpressing HEK293 cells and demonstrated that intracellular zTGs2 plays a very comparable protective/damaging role in the apoptotic process, as hTG2. Overall, our results suggest that zTGs2 proteins behave very similarly to the human ortholog and pave the way for future in vivo studies of TG2 functions in zebrafish.

## 1. Introduction

Transglutaminase 2 (TG2) is the most widely distributed member of the transglutaminase family whose common feature is Ca^2+^-dependent catalysis of protein cross-linking [1]. Its gene encodes a 77-kDa protein consisting of four domains: an N-terminal β-sandwich responsible for the cell adhesion functions of TG2, a catalytic core containing the catalytic Cys-His-Asp triad responsible for the cross-linking activity, and two C-terminal β-barrel domains important for regulating TG2 activity and for its involvement in cell signaling [2]. The enzymatic functions of TG2 are regulated by a conformational change that shifts the protein from an elongated and active form favored by high Ca^2+^ concentrations to a closed, inactive form promoted by GTP/GDP binding, in which the β-barrels are folded over the catalytic core, shielding the active site, and inhibiting transamidase activity [3,4]. After binding to GTP, TG2 exerts GTPase activity and mediates GPCRs signaling as an atypical G protein [5,6,7]. TG2 is widely distributed in tissues and localizes in both the intracellular and extracellular environments, where it serves distinct functions [8,9,10,11,12].

In the extracellular matrix (ECM), TG2 mediates matrix remodeling by cross-linking ECM components such as collagen and fibronectin (FN) and supports cell adhesion functions by acting as a bridge between the ECM and cells [13,14]. Specifically, TG2 can form complexes with fibronectin and bind to syndecan-4 on the cell surface, which then activates inside-out signaling of α5β4-integrins and ultimately leads to cell adhesion and spreading through the formation of focal adhesions, but also pro-survival signaling through MAPK activation and deposition of new fibronectin fibrils in the matrix [15,16]. In the intracellular compartment, TG2 may be localized to the inner surface of the plasma membrane, where it functions as a G protein that mediates signaling upon activation of G protein-coupled receptors (GPCRs), but also in the cytosol and in various organelles such as the mitochondria and nucleus, with effects on various processes such as transcriptional regulation, stress response, and apoptosis [11,17,18]. 

Due to its diverse functions and expression in different cell types, TG2 is involved in many physiological and pathological processes, such as bone development, wound healing, angiogenesis, as well as fibrosis, celiac disease, and cancer [19,20,21]. However, its actual role in these biological processes is controversial, as TG2 has often shown opposite effects in different processes depending on localization, cell type, or experimental conditions.

The zebrafish is a proven animal model for studying vertebrate biology and recapitulating various human pathologies [22,23]. It offers several advantages, including high genetic similarity to humans and the ability to produce hundreds of externally fertilized and optically clear embryos each week, characterized by rapid development and ease of manipulation [24]. Very little is known about the zebrafish transglutaminase 2 orthologs from both genetic and functional perspectives. The TG2 orthologs of the zebrafish were first described by Deasey and coworkers [25]. They identified three genes named *zTG2b, zTG2c*, and *zTG2-12*, with *zTG2b* identified as the isoform with the most ubiquitous expression in all tissues and during all developmental stages, while the expression of the other two decreased from the larval stage to the adult stage. From a functional perspective, they showed that mineralization of newly formed vertebrae was significantly reduced in zebrafish embryos when treated with a TGs inhibitor. In a recent publication, the zebrafish was used as a tool to study the relationship between TG2 and the Wnt/β-catenin signaling pathway by knocking down the three genes encoding zebrafish TGs2 orthologs [26]. Visible morphological changes, loss of pigmentation, suppression of the Wnt/β-catenin pathway, and increased mortality were observed in embryos in which the *zTG2b* isoform was downregulated, whereas downregulation of the other two zTGs2 genes had no direct effect on embryo viability and morphology, suggesting that *zTG2b* may be the TG2 ortholog in zebrafish that plays an important role in embryonic development. The transglutaminase 2 ortholog protein was also recently studied in another fish model, the medaka fish, where the protein was found to be enzymatically active and its deletion resulted in a delay in the movements of the fish [27,28]. In this study, our goal was to further investigate the genetic and functional properties of TG2 orthologs in zebrafish to provide the basis for using zebrafish as a valid tool for in vivo characterization of TG2 functions. To this end, three zebrafish TG2 orthologs were studied in detail, and we show that all orthologs are highly similar to human TG2 from a structural, biochemical, and functional perspective by demonstrating that zTGs2 can mediate Ca^2+^-dependent transamidation activity both as recombinant proteins and as heterologous proteins expressed in HEK293T cells. In addition, we report that zTGs2, like hTG2, can mediate RGD-independent cell adhesion in the extracellular environment and an anti/pro-apoptotic role in the intracellular compartment. These results confirm that many functions that TG2 exerts in humans are conserved in zebrafish orthologs.

## 2. Results

### 2.1. Evolutionary History of Zebrafish Transglutaminase 2

The orthologous TG genes in the zebrafish genome were originally described by Deasey and coworkers more than 10 years ago when only a handful of ray-finned fish genomes were available [25]. We used up-to-date zebrafish genomic information to perform a new phylogenetic maximum likelihood analysis (ML) to clarify the evolutionary relationships between zebrafish and human orthologs [29,30]. The data confirmed that the zebrafish TG gene family comprises 14 genes. Six of these clustered with high confidence (bootstrap support = 76) with human *TGM1* and three with *FXIIIA*. The zebrafish genome encodes three sequences that belong with high confidence (bootstrap support = 85) to the same monophyletic clade of human *TGM2* and *EPB42* genes, highlighted by a blue box in Figure 1. Two of these, designated *tgm2a* and *tgm2b* by current nomenclature, can be considered as *bona fide* orthologs of the human gene (red box), whereas the third zebrafish sequence, named *tgm2l*, was more distantly related.

The presence of a *tg2m2l* gene in the genomes of most (61 out of 64) Ensembl reference bony fish species, including the basal spotted gar *Lepisosteus aculeatus* [31] supports an ancient evolutionary origin for this gene, predating the radiation of teleosts. In detail, the birth of *tg2m2l* can be traced back before the split between Actinopterygii and Sarcopterygii (as suggested by the presence of *tgm2l* in cartilaginous fishes), but after the split between jawed vertebrates and Agnatha, which only have a single-copy *TGM2*/*tgm2l* gene. Interestingly, the presence of *tgm2l* in coelacanth, paired with its lack in all extant land vertebrates, further indicates that this gene was lost in the latest common ancestor of all tetrapods. On the other hand, while zebrafish displays the presence of two distinct *tgm2* paralogs, the number of gene copies was highly variable in bony fishes, as evidenced by the frequent loss of orthologs in several species, and by the presence of additional paralogs in others. In detail, 27% of species (17 out of 64), including medaka (Figure 1) lacked *tgm2a* orthologs and 14% (9 out of 64) lacked *tgm2b* orthologs. We further assessed the presence of gene copy number variation (CNV) in zebrafish, analyzing the publicly available genomic sequences of nine strains, and did not detect any evidence supporting the presence of additional *tgm2a/b* and *tgm2l* gene copies. 

### 2.2. Gene Expression Analysis of Zebrafish Transglutaminase 2 Genes

Gene expression of zebrafish TGM2 orthologs has been previously studied at various stages, particularly at post fertilization day (dpf) 4 and 13 and in adults [25]. We decided to investigate expression at earlier time points in zebrafish embryonic development. cDNA was obtained from whole embryos and analyzed at 1, 3, and 5 dpf. Results showed that all three genes were expressed throughout early embryonic development. *Tgm2a* was the gene with the highest expression among the three genes, but its transcription level decreased during the first 5 days of development. A different expression profile was found for *tgm2b*, whose expression level did not change significantly during the first 5 days following fertilization. Although the expression of *tgm2l* was very low and progressively decreased during the first days of development (Figure 2A), this evidence of transcription, together with the conservation of the sequence across a broad taxonomic range, strongly supports the identification of *tgml2* as a functional gene. In parallel with gene expression, we performed two different assays to test whether the cross-linking enzymatic activity was present. First, we collected embryos at the same time points on days 1, 3, and 5, and tested the whole lysate for enzymatic activity by measuring the incorporation of pentylamine-biotin into the substrate N,N-dimethylcasein (DMC) coated on a solid-phase surface. Figure 2B shows that the activity was present on day 1, increased on day 3, and remained stable until day 5. The presence of EDTA, which inhibits calcium-dependent transglutaminase activity, completely abolished the reactivity of the assay, suggesting the presence of activity associated with TG. The second assay detects transglutaminase activity by using a fluorescent TG2 primary amine substrate to directly observe cross-linking mediated by the enzyme in vivo in zebrafish embryos. At 2 dpf, Tg(fli1:EGFP) embryos were microinjected with the red fluorescent TG2 competitive amine substrate tetramethylrodamine (TAMRA)-cadaverine into the yolk sac and observed 3 days after injection after extensive washes (Figure 2C). As shown in Figure 2D, labeling was visible in the trunk region of the embryos, with fluorescent cadaverine recorded in both the dorsal and ventral regions, particularly in the vasculature visible for GFP expression.

These data demonstrate that the zebrafish *TGM2* orthologs are expressed throughout embryonic development and are likely expressed as functional proteins as TG crosslinking activity can be detected both in vitro and in vivo in zebrafish embryos.

### 2.3. Zebrafish TGs2 Proteins Sequence and Structure Analysis

We obtained cDNA from whole zebrafish embryos and cloned the three zebrafish orthologs of human transglutaminase 2 using specific primers. The cloned genes were sequenced to determine the amino acid sequences for the three proteins (zTGs2), which we named zTG2a, zTG2b, and zTG2like. Both the DNA and amino acid sequences were then used to examine sequence divergence compared with the zebrafish reference strains. Interestingly, we found that the zTGs2 sequences diverged significantly between the strains studied. Whereas the sequence of cloned zTG2a is identical to that of the strain SAT, the sequence of cloned zTG2like is identical to that of the Tuebingen reference strain, and that of cloned zTG2b shows a different degree of divergence from the sequences of all other strains examined (Appendix A). Our results suggest that the observed differences are likely due to allelic variants that occur in different zebrafish populations and are not strictly associated with a particular strain. Subsequently, we aligned the three zTGs2 protein sequences with the human transglutaminases TG1-7 and FXIIIa (Figure 3A). We found that the protein sequence reflected the evolutionary relationships between the coding genes shown in the cladogram in Figure 1. zTG2b showed a higher degree of similarity with the human TG2 enzyme, with a homology of 68%. When compared with other members of the human TGs protein family, homology ranged from 43% for hTG1 to 56% for hTG3. Similar results were obtained for zTG2a, which has 65% homology with hTG2 compared with 42% for hTG1. TG2like was the isoform with the lowest similarity, achieving only 58% homology with hTG2. Since reliable structure prediction tools have recently become available, we decided to model the three-dimensional structure of TG2 in zebrafish. The 3D structures of the cloned proteins were modeled using Alphafold via Colabfold and later used for pairwise structure alignment with human TG2 [32,33,34]. As expected, the proteins show the typical structural organization into four domains: an N-terminal β-sandwich, a catalytic core, and two C-terminal β-barrels. This was confirmed by a very high TM-score with values ranging from 0.96 to 0.97. Root mean square deviation (RMSD) was then calculated between the aligned pairs and the values were consistent with sequence homology, with zTG2b having an overall RMSD of 1.1 Å, followed by zTG2a with 1.66 and zTG2like with 1.75 Å, reflecting lower homology and greater evolutionary distance.

### 2.4. Recombinant zTGs2 Proteins Have a Ca^2+^-Dependent Activity

Prior to the biochemical characterization of the three zTGs2 as recombinant proteins for their enzymatic and non-enzymatic functions, a multiple sequence alignment was performed between the zebrafish and human TG2 to determine whether the core catalytic residues responsible for the transamidation activity of the transglutaminases were conserved. As shown in Figure 4A, there is a high degree of homology between all residues in the three regions involved, and in particular, the Cys-His-Asp catalytic triad is conserved in all three zebrafish proteins, as are the relative positions between these three residues in the 3D structure of the protein, as shown in Figure 4B. The zTGs2 genes were then cloned into an expression vector and the recombinant proteins were produced in bacteria as a C-terminal His-tag fusion. Small-scale colony cultures were first performed at various growth temperatures (from 20 °C to 37 °C) and IPTG induction conditions (from 0.1 to 1 mM), to optimize protein production because although all proteins were expressed at significant levels, the soluble fraction was limited. The soluble fraction produced by the selected optimized production protocol was subjected to affinity purification by IMAC, and protein integrity and purity were confirmed by SDS-PAGE and Western blot assays (Appendix A). In an initial assay for enzymatic activity, the purified enzymes were tested for their ability to incorporate pentylamine-biotin into the substrate N,N-dimethylcasein in solution. All three zebrafish TGs2 as well as the human enzyme were able to mediate the transamidation reaction in the presence of Ca^2+^, whereas no enzymatic activity was detected in the absence of calcium ions and in the presence of the calcium chelating agent EDTA (Appendix A). This test does not show clear differences between the signals obtained with the zebrafish proteins. To obtain a more quantitative evaluation, we performed the transamidation reaction against the DMC substrate coated on a microplate, testing all proteins at increasing concentrations and using human TG2 as a reference. Under these experimental conditions, it was confirmed that the activity was concentration-dependent for all tested proteins, and differences were observed between the tested proteins. Human TG2 showed higher signal at all concentrations, and zTG2a and zTG2like overlapped and showed higher signal compared with zTG2b at the same concentration (Figure 4C). Interestingly, these data clearly confirm that the zTG2like protein previously identified as “predicted” is not only expressed during embryonic development but is also fully functional in transamidation activity.

We also investigated whether zTG2 proteins can be inhibited by hTG2-specific inhibitors. Specifically, we tested sensitivity to three known inhibitors: ZDON (Figure 4D), R281 [35] (Figure 4E), and 1-155 [36] (Figure 4F), each at increasing concentrations starting at 0.25 μM and extending to 25 μM. Each enzyme activity was inhibited to varying degrees depending on the concentration and type of inhibitor used. At the higher concentration, we observed approximately 50% inhibition of the 3 zTGs2 by ZDON, whereas both R281 and 1-155 resulted in almost complete inhibition of zTGs2 activity, with the sole exception of zTG2a, which was less sensitive to the inhibitory effect of 1-155 than zTG2b and zTG2like. This difference in reactivity toward the 1-155 molecule was even more evident at 2.5 μM, where zTG2b and zTG2like were 80% inhibited, whereas the activity of zTG2a was reduced by only 20%. At 2.5 μM, the inhibitory effect of ZDON was very small compared with the other two inhibitors. 

Overall, we succeeded in cloning, producing, and purifying all three zTGs2 as soluble and functional proteins. Furthermore, we have shown that active site conservation correlates positively with conserved transamidation activity, which could be inhibited by TG2 selective inhibitor compounds.

### 2.5. The zTGs2 Proteins Have a Ca^2+^-Dependent Activity but Are Not Inhibited by GTP

The transamidation activity of mammalian TG2 is calcium-dependent, as high Ca^2+^ concentrations cause a large conformational change leading to a catalytically active conformation of the enzyme. Conversely, low Ca^2+^ levels and the presence of GTP/GDP promote the conformational change to a closed and inactive form. We decided to investigate whether the enzymatic activity of zTGs is regulated by the presence of calcium. First, we aligned the zebrafish TGs2 sequence with that of the human enzyme and examined the conservation of the five putative calcium binding sites identified by Kiraly and colleagues (Figure 5A) [37]. Key residues responsible for interaction with Ca^2+^ are largely conserved in zebrafish TGs2 proteins. High homology was particularly evident in regions S2, S4, and S5, where 12 of 15 relevant residues were conserved for zTG2a and 13 for zTg2b and zTG2like, respectively. To confirm the calcium dependence of transamidation activity, we tested the recombinant zTGs2 proteins at increasing concentrations of CaCl_2_. All three proteins exhibited an increase in enzymatic activity that was directly related to calcium concentration (Figure 5B). Because Ca^2+^ and GTP/GDP have an opposite effect on activity, we investigated whether zebrafish TGs2 might be sensitive to the presence of GTP. First, we examined the protein sequences to determine whether the zTGs2 exhibited homology to the putative regions responsible for GTP binding of the human enzyme. In contrast to the calcium-binding sites, partial homology was present only in the sequence of zTG2a, whereas there was no conservation of the putative key residues for zTG2b and zTG2like (Figure 5C). These data were in line with the GTP inhibition assay which showed that zTG2a is the only zebrafish TG2 whose activity is affected by the presence of GTP, although the inhibitory effect of GTP was only one third of that observed for the human enzyme (Figure 5D).

In summary, we have shown that all three zTG2 proteins possess Ca^2+^ dependent catalytic activity but, unlike the human enzyme, do not exhibit sensitivity to the inhibitory effects of GTP.

### 2.6. Intracellular zTGs2 Proteins in HEK293 Cells Mediate Different Effects against Cell Death in a Stimulus-Dependent Manner

Since enzymatic characterization in vitro has shown that zTGs2s enzymes largely mimic the human ortholog, we wanted to test whether this is also the case under physiological conditions. HEK293T cells were chosen as a cell model because they are known to express low levels, if any, of endogenous hTG2. Cells were transfected with a eukaryotic expression vector encoding each of the zTGs2 fused to an SV5 tag and with hTG2 as a control. At 48 h after transfection, protein expression was detected to varying degrees in the cell extract. Therefore, to avoid etherogenicity, stably transfected cultures were selected for each protein. Protein expression and cell population homogeneity were verified by both citofluorimetric analysis and WB on cell lysates, which confirmed that all cell lines expressed comparable amounts of protein (see Figure 6A,B). Later, transamidating activity in the cell lysate was examined, revealing a negligible signal for wild-type cells and the presence of active proteins with comparable signals for all transfected cell lines (Figure 6C). To investigate whether the enzyme plays a functional role, we first focused on apoptosis because there is clear evidence that TG2 is involved in this process and either promotes or attenuates apoptosis. Since these effects have been shown to depend on the type of apoptosis trigger, we used two different stimuli and evaluated the results.

First, we worked with a hydrogen peroxide stimulus. The optimal H_2_O_2_ concentration and exposure time were first determined on non-transfected cells, and finally, stable clones of TGs2-expressing cells and wild-type HEK293T cells were treated with 0.4 mM H_2_O_2_ for 2 h. Cell death was determined by flow cytometry using Annexin V and propidium iodide staining (Figure 6D). First, we found that apoptosis in untreated cell cultures did not differ significantly between WT and transfected cells. In contrast, when cells were treated with H_2_O_2_ to induce apoptosis, there was a clear effect attributable to the presence of TG2 proteins in the cells. Whereas in the WT sample, the percentage of cells that underwent apoptosis after the stimulus increased 7.1-fold, there was a protective effect in TG2-expressing cells, with increases ranging from only a 1.6-fold for zTg2like to 3.2-fold for zTG2A (Figure 6E). Caspase3 activity was measured as an indicator of apoptosis, and data confirmed that exposure to hydrogen peroxide caused higher activity in nontransfected cells compared with TG2-expressing cells (Figure 6F). 

As a second stimulus, TGs2-expressing cells and wild-type HEK293T cells were treated with 5 μM thapsigargin in serum-free medium for 24 h. In this case, exposure to thapsigargin resulted in a 1.7-fold increase in the percentage of apoptotic cells in the WT cell compared with untreated control cells. In contrast to previous results observed with exposure to H_2_O_2_, expression of zTGs2 and hTG2 appeared to enhance the apoptotic process induced by thapsigargin. In particular, the cells expressing zTG2a and zTg2b became more sensitive and showed a 2.7-fold increase in the number of apoptotic cells after treatment (Figure 6G). In this case, caspase-3 activity was present but did not differ significantly between transfected cells (Figure 6I).

These results demonstrate that all zTGs2 can be expressed and are active in the intracellular environment. Moreover, in response to apoptotic stimuli, zTGs2 exert a specific role that is either protective or sensitizing, depending on the stimulus, and is similar to that of human TG2.

### 2.7. The zTGs2 Proteins Mediate Can Mediate RGD-Independent Cell Adhesion and Spreading

TG2 plays an important role not only in the intracellular environment but also in the extracellular. TG2 in the extracellular matrix can rescue RGD-independent cell adhesion by forming complexes with matrix fibronectin (FN) through its binding to heparan sulfate proteoglycans (HSPGs) on the cell surface. To understand whether TGs2 in zebrafish has a similar function to TG2 in mammals, we first examined the conservation of putative key residues for interaction with HSPGs by performing multiple protein sequence alignment of the different regions, identified as potential heparan sulfate binding sites. Different levels of conservation were observed for the three zebrafish TGs2, with zTG2a and zTG2b showing significant discrepancies in these regions, despite being the two isoforms with the higher overall homology to hTG2 (Appendix A). The two proteins were analyzed to investigate their contribution to mediating cell adhesion and spreading of Human Foreskin Dermal fibroblasts (HFDFs). First, we found that when human or zebrafish TGs2 were used as a single coating substrate on the culture plate, no cell adhesion was detectable, confirming that they are not able to mediate cell adhesion alone. In a second step, we measured adhesion and spreading when a TG2-fibronectin (FN) matrix heterocomplex was used as substrate. In this case, we observed significant cell adhesion (Figure 7A,B) and spreading (Figure 7B,C) for all four protein complexes, with values slightly higher than when FN alone was used. Finally, we tested the effect of incubating the cells with a specific RGD-containing peptide. For fibronectin alone, incubation with the peptide resulted in a 50% reduction in cell adhesion (Figure 7A,B) and a 70% reduction in cell spreading (Figure 7B,C) compared with the control condition without peptide, as expected. In contrast, we observed that cells incubated with the inhibitory RGD peptide and seeded onto a zTGs2 FN complex fully restored both attachment and spreading. In this regard, the effects of the two zebrafish proteins were not different from those observed for hTG2. Cells were also stained with FITC-labeled phalloidin, and the actin cytoskeleton of seeded fibroblasts was imaged by epifluorescence microscopy. As shown in Figure 7D, fibroblasts treated with the RGD peptides show a small and rounded morphology when seeded on fibronectin, whereas seeding of the zTGs2 FN complexes leads to reorganization of the actin cytoskeleton with the formation of stress fibers and protruding structures.

These data demonstrate that TGs2 in zebrafish can exert a cell adhesion function similar to that previously described for the human ortholog and likely acts as a bridge between cell surface receptors and matrix fibronectin.

## 3. Discussion

Tissue transglutaminase (tTG or TG2) is widely distributed in the human body and is highly expressed in endothelial cells and myofibroblasts. However, its functions have not been fully elucidated [38]. Although it is clear that TG2 is involved in physiological processes such as wound healing, angiogenesis, and tissue remodeling, as well as pathological conditions such as cancer, fibrosis, and celiac disease, the extent to which this protein contributes to each of these phenomena is far from clear [38,39,40,41,42]. With the goal of finding a suitable model to better characterize the physiopathological functions of TG2, we decided to describe the TG2 ortholog in zebrafish and investigate whether the animal could be considered an ortholog of humans. Until now, little was known or described in the literature about the transglutaminase gene family in *Danio rerio*. Since the initial description of the zebrafish gene family TG by Deasey and coworkers, this topic has been addressed in only a few publications, with very little additional information on the functional properties of TG in this organism [25,26,43]. To obtain a more detailed and up-to-date description of the TG family in zebrafish, we performed a new phylogenetic analysis, supported by a comparative investigation of the 64 currently available ENSEMBL high-quality references genomes of ray-finned fishes. Our data confirm that the zebrafish gene family TG contains 14 genes, including five orthologs of *TGM1*, three for *F13A1*, and three for *TGM2* (Figure 1). All zebrafish TG genes encode proteins that, despite significant primary sequence diversity and some length variation, are characterized by the canonical domain architecture typical of TG proteins: an N-terminal β-sandwich domain followed by a central catalytic core domain comprising the catalytic triad and two C-terminal β-barrel domains [44].

Of the three zebrafish sequences included in the *TGM2* clade, *tgm2a*, and *tgm2b* can be considered as orthologs of the human *tgm2* gene, which underwent duplication (as in most other fish species), whereas *tgm2l* is evolutionarily more ancient, as its origins were inferred to predate the radiation of teleosts. We demonstrated that the three genes are expressed during zebrafish embryonic development, with *tgm2a* and *tgm2b* expressed at significantly higher levels than *tgm2l*. However, the conservation of *tgm2l* in all bony fishes supports a biological role for the zTG2like protein, even though its function is restricted to these organisms since the gene was lost during the water-to-land transition. In addition, the functional importance of this gene in teleosts is also supported by the non-negligible expression in various adult tissues observed in the analysis of RNA-seq data. The two paralogs, *tgm2a*, and *tgm2b*, likely encode proteins that may perform similar functions in bony fishes, as suggested by the frequent loss of the ortholog of either gene in other species. This may indicate some degree of functional overlap between the two paralogs, with the loss of one of the two genes likely compensated for by the presence of the second copy. Simultaneous loss of both gene copies is extremely rare in evolutionary history, as a single species, *Oryzias sinensis*, lacks a *TGM2* ortholog (but still possesses a TGM2L gene). Because antibodies are not available to accurately determine the expression of zTGs2 proteins in zebrafish embryos, we obtained some indirect evidence for the functional expression of TG proteins. First, we demonstrated the presence of Ca^2+^-dependent TG activity in embryo lysates at three time points during embryonic development, and then we showed how a known TG2 competitive amine substrate such as cadaverine, is actively incorporated into living embryos after microinjection. This in situ incorporation could be attributed to the presence of functional proteins with TG activity capable of cross-linking the substrate.

Before proceeding with biochemical and biological functional characterization of the zTg2s proteins, we predicted their structure using ColabFold [34], a powerful software that integrates AlphaFold’s advanced AI system. As expected, all three models contain large regions of predicted Local Distance Difference Test (lDDT) scores > 90, indicating very specific structure prediction. When pairwise aligning each model with the human TG2, the RMSD values were extremely low with a TM-score reaching 0.97. In general, the structures showed the typical conserved domain architecture comprising an N-terminal β-sandwich domain followed by a central catalytic core domain and two C-terminal β-barrel domains. The encoded proteins exhibit high conservation of the core catalytic regions responsible for the transamidation activity of transglutaminases, and this high degree of homology is reflected in their ability to catalyse the transamidation reaction. Although the position of the triad in the protein sequence is slightly shifted in zebrafish proteins compared with the human (Cys277-His335-Asp358), the relative position between the three residues is conserved, suggesting that the spatial orientation of these residues may be conserved in the catalytic core. 

Not only are the residues directly involved in catalytic activity conserved, but also many of the residues responsible for the formation of Ca^2+^-binding sites essential for the transamidation reaction of hTG2 are conserved in zTGs2, whose transamidation activity is indeed Ca^2+^-dependent. The transamidation activity of zTGs2 is not sensitive to the inhibitory effect of GTP as is the human ortholog. The experimental data are consistent with the results of sequence analysis of the residues responsible for GTP binding. According to Han et al., hTG2 has a GTP-binding cleft formed by Phe174, Ser482, Met483, Arg476, Arg478, Arg580, and Tyr583 that is well conserved in mammals but not in birds, frogs, and fishes [45]. Studies on the TG2 ortholog of medaka fish revealed that this stretch of residues is not conserved in this organism [24]. Moreover, they showed that GTP could not inhibit the transamidation activity of medaka TG2 in ELISA because these key amino acid residues were absent in the sequence, as we observed for zTG2b and zTG2like. According to our data, only zTG2a appears to be affected by GTP, with inhibition reaching only one third of that observed for hTG2. Although only a few residues are conserved, other residues retain the physicochemical properties of the substituted amino acid (two lysine instead of arginine and an aspartate instead of a glutamate). It is plausible that zTG2a could bind GTP to a small extent. Structural studies should be performed to determine whether GTP does not bind to the zTGs2s at all, since they lack a nucleotide-binding cleft, and whether the zTGs2s undergo a conformational change like the human enzyme. 

In addition to the endogenous regulatory mechanism, we also investigated three known hTG2-specific inhibitor molecules whose IC50 values for the human enzyme are in the low nanomolar to micromolar range. R281 is a peptidic water soluble inhibitor that is cell impermeable and able to differentiate between TG2 and Factor XIIIa; ZDON is a commercial TG2 inhibitor that is TG2-specific and displays less effect on inhibiting TG1 and TG3 [20]; and 1-155 is a highly potent TG2-specific cell-permeable inhibitor that has shown effectiveness in inhabiting fibrosis and cancer progression [21,36,39,42]. For all three inhibitors, we observed significant inhibition of zTGs2 activity at different concentrations, which were 2.5 μM for R281 and 1-155 and 25 μM for ZDON. Although the IC50 values of the inhibitors are much lower for the human enzyme, the results confirm that the structure of the catalytic core of zTG2 is very similar to that of hTG2.

Multiple roles in programmed cell death have been described for hTG2, with studies suggesting both pro- and anti-apoptotic roles, depending on the apoptotic stimuli, TG2 localization, and cell type [12,46,47,48,49]. Studies in human embryonic kidney cells (HEK293) have shown that overexpression of TG2 has anti-apoptotic effects. Specifically, hTG2 has been shown to impair the mitochondria-mediated apoptotic pathway by downregulating the expression of Bax, thereby decreasing mitochondrial permeability. This leads to a decreased mitochondrial release of cytochrome c and ultimately downregulation of caspase-3 and caspase-9 activation, thereby inhibiting apoptosis [50]. These anti-apoptotic functions of TG2 have also been confirmed in renal cell carcinoma (RCC), where overexpression of TG2 leads to enhanced autophagy with protective anti-apoptotic effects [51]. Specifically, TG2 has been shown to mediate cross-linking of p53 in autophagosomes, leading to the depletion of p53 and preventing p53-induced cell death signaling in RCC. However, it has also been shown that overexpression of TG2 in HEK293 cells has pro-apoptotic effects when exposed to thapsigargin, an inhibitor of endoplasmic reticulum Ca^2+^-ATPase, leading to increased intracellular calcium levels and induces apoptotic cell death in various cell models [46]. In HEK293 cells transfected with hTG2, thapsigargin has been shown to increase cytosolic TG2 activity, which correlates with increased caspase-3 activation compared to untransfected control cells.

Based on these observations, we generated stable transfected HEK293 cells overexpressing functional zTGs2 to test whether zebrafish proteins could mediate some of the effects described during cell death triggered by various stimuli. When cell apoptosis was triggered by toxic concentrations of hydrogen peroxide, we showed that expression of any of the three zTGs2 effectively reduced the percentage of cells undergoing cell death. This reduction in cell death was statistically significant and comparable to that observed in cells overexpressing hTG2. Opposite results were observed when cells were exposed to a different stimulus such as thapsigargin. In this case, an increase in the percentage of apoptotic cells overexpressing zTG2s proteins was observed, which is consistent with the observations of Milakovic and coworkers who reported increased caspase-3 activity in TG2-overexpressing cells after thapsigargin treatment [46]. 

TG2 plays an important role as a cell adhesion molecule in the extracellular environment. It acts as a linker between fibronectin and syndecan-4 with an RGD-independent mechanism that is critical to prevent loss of adhesion during matrix turnover and tissue injury and promotes pro-survival signaling [15,16]. We demonstrated that this function is conserved in zebrafish, as zTGs2 can mediate cell adhesion on HFDFs treated with RGD peptides and promote both attachment and spreading with reorganization of the actin cytoskeleton. These observations suggest that zebrafish TGs2 may bind to heparan sulfate proteoglycans on the cell surface and activate the same integrins for inside-out signaling. Actin cytoskeleton staining showed that the seeding of RGD-treated cells on zTGs2-fibronectin complexes led to the formation of actin stress fibers and promoted cell attachment and spreading. Human TG2-fibronectin complexes binding to syndecan-4 lead to focal adhesion formation through activation of focal adhesion kinase (FAK) mediated by α5β4-integrin signaling, so it is plausible that zTGs2 could exert its cell adhesion function by the same or similar mechanism.

The zebrafish is an increasingly successful model for translational research into various physiological and pathophysiological conditions in humans, which also allows the detailed study of the functions of several orthologous proteins and enzymes. By demonstrating that the transglutaminase enzymes present in this animal model reflect specific features of the human transglutaminase-2 enzyme, the zebrafish model may be instrumental in clarifying several as yet unexplained functions of this enzyme.

## 4. Materials and Methods

### 4.1. Genetic Analysis of TG2 Orthologues in the Zebrafish Genome

The coding sequences (CDS) of all human genes belonging to the transglutaminase gene family (i.e., TGM1-7, FXIIIA and EPB42) were retrieved from RefSeq. Danio rerio orthologs, obtained from Refseq Annotation Release 104 and Ensembl Release 109 (February 2023) [29,30]. were corrected, whenever necessary, through their alignment with the available de novo assembled zebrafish transcriptomes (NCBI accession IDs: GFIL00000000.1, GDQQ00000000.1 and GDQH00000000.1). A few additional fish sequences related to TGM2 (i.e., those from the medaka *Oryzias latipes* and the spotted gar *Lepisosteus oculatus*) were obtained from Ensembl.

All deduced amino acid sequences were aligned with MUSCLE v.5.1 and the resulting multiple sequence alignment (MSA) was processed with Gblocks to remove the noninformative regions that were too divergent or corresponding to significant small insertions and deletions (indels) [52,53]. The MSA was used as an input for a Maximum Likelihood (ML) phylogenetic analysis with IQ-TREE v. 1.6.12 using 100 ultrafast bootstrap replicates [54,55]. The analysis was run based on a LG+I+G4 model of molecular evolution, which was selected as the best-fitting for the input MSA with ModelFinder, according to the Bayesian Information Criterion. Phylogeny was graphically represented as a midpoint-rooted phylogram using iTOL v.6 [56,57,58].

Orthology/paralogy relationships for the *tgm2a*, *tgm2ab* and *tgm2l* genes were obtained from Ensembl Release 109 [30]. The presence of the three genes was investigated though the analysis of the genomic RefSeq sequences (Tuebingen strain) and those retrieved from eight additional strains: CG2 (GCA_001483285.2), T5D (GCA_018400075.1), CB (GCA_903798165.1), NA (GCA_903798175.1), SAT (GCA_020064045.1), CG1 (GCA_025582565.1), AB (GCA_020184715.1) and AB2 (AB2GCA_025582595.1).

### 4.2. Zebrafish Embryos Handling and Maintenance

Fertilized zebrafish eggs were placed in E3 Medium (5 mM NaCl, 0.17 mM KCl, 0.33 mM CaCl_2_, 0.33 mM MgSO_4_) supplemented with methylene blue 0.5% and incubated at 28 °C. At 24 h post-fertilization (hpf), the eggs were dechorionated with Pronase 1 mg/mL for 5 min at room temperature (RT). Following a wash in E3 Medium, the embryos were placed in E3 Medium supplemented with Phenylthiourea (PTU) 0.2 mM (final concentration) to prevent melanogenesis and thus the pigmentation of the larvae. Zebrafish embryos were handled according to standard rules and procedures for animal wellness “https://zfin.org accessed on 01 June 2022”. All experimental procedures involving animals were executed after Ministerial Approval 1FF80.N.LWE.

### 4.3. Gene Expression Study of the Three Zebrafish TG2 Genes during the Embryonic Development

The expression of the three zebrafish TG2 (zTGs2) genes was studied by Quantitative Real-Time PCR (RT-qPCR) at 1, 3, and 5 days post-fertilization (dpf). 

Zebrafish embryos were euthanized with 0.2% Tricaine. Total RNA was extracted from experimental groups of 20–30 embryos with the PureLink™ RNA Mini Kit (Invitrogen^TM^, Carlsbad, CA, USA, cat#12183020) after embryos’ lysis with TRIzol^®^ reagent (Life technologies, Carlsbad, CA, USA, cat#15596), following the manufacturer’s instructions.

Primers pairs were designed using the open source Primer3 program version 4.1.0 (https://primer3.ut.ee/) to map on different exons, to generate an amplicon of about 250–300 bp, and to have a melting temperature (tm) of about 60 °C. Two housekeeping genes were selected as normalizers, LSM12 homolog b (*lsm12b*) and beta-actin (*β-actin*). The primer pair of lsm12b was previously validated by the work of Yu Hu and co-workers, whereas the primer pair of β-actin was previously validated by the work of Tang et co-workers [59,60]. All the primers used for the gene expression analysis are listed in Table 1.

RNA was extracted as previously described from 1, 3, and 5 days post fertilization (dpf) zebrafish embryos. After treatment with ezDNase (Invitrogen^TM^, cat#11766051), 5 µg of total RNA was reverse transcribed to produce cDNA using SuperScriptTM IV Reverse Transcriptase (Invitrogen^TM^, cat#18090200). cDNA was diluted fivefold and used as a template to perform a quantitative Real Time PCR (qRT-PCR) with iQ™ SYBR^®^ Green Supermix (Bio-Rad, Hercules, CA, USA, cat#1708880) following the manufacturer’s instructions in a CFC96 thermocycler (Bio-Rad). The relative amounts of the transcript were calculated with the comparative ΔΔCt method using β-actin and lsm12b as normalizers.

### 4.4. Zebrafish Embryos Lysates Transamidation Activity Assay

Zebrafish embryos protein lysates were obtained starting from 30 embryos at 1, 3, and 5 dpf. The embryos were lysed first by pipetting with an insulin syringe in 600 μL of lysis buffer (50 mM Tris-HCl, pH8, 150 mM NaCl, 1% Triton X100), and then by sonication. After 15 min of centrifugation at 14,000× *g* at 4 °C, the clear lysates were quantified with Bradford reagent. ELISA wells were coated O/N at 4 °C for 16 h with 120 µL of 15 µg/mL N,N′-dimethyl casein (DMC) in PBS. After one wash with PBS, 100 µL of reaction mix (100 mM Tris-HCl, pH 8.0, 0.2 mM EZ-Link™ Pentylamine-Biotin from Thermo Scientific cat#21345, 10 mM DTT, 5 mM CaCl_2_) with 20 μg of zebrafish embryos protein lysates were added in each well and the plate was incubated for 2 h at 37 °C. Following three washes with PBS-Tween 20 0.1% and three with PBS, the plate was incubated for 1 h at 37 °C with 100 µL per well of Pierce™ High Sensitivity Streptavidin-HRP diluted in a solution of 1% BSA in PBS in volumetric ratio 1:200. After washes, the colorimetric reaction was developed by adding 100 µL per well of TMB and stopped after 5 min by adding 50 µL per well of 2.5 mM H_2_SO_4_. The absorbance was measured at 450 nm using a microplate reader (TECAN^®^, Männedorf, Switzerland).

### 4.5. Fluorescent Cadaverine Incorporation in Live Zebrafish Embryos

2 dpf Tg(fli1:EGFP) zebrafish embryos were anesthetized with 0.02% Tricaine and microinjected in the yolk sac with 2.3 nL of 400 μM N-(Tetramethylrhodaminyl)cadaverine (TAMRA-cadaverine, Zedira cat#R001). The embryos were placed in E3 Medium supplemented with Phenylthiourea (PTU) 0.2 mM (final concentration) to prevent melanogenesis. Three days after the injection, epifluorescence images of live anesthetized embryos were captured with a Nikon Eclipse Ti-E live system at 20× magnification.

### 4.6. Molecular Cloning of Zebrafish Transglutaminase 2 Sequence into Bacterial Expression Vector

The three zebrafish TG2 coding sequences have been cloned in the pET30a(+) bacterial expression vector in order to produce the three TGs2 as His6-tagged recombinant proteins. Zebrafish TGs sequences were PCR-amplified with Phusion™ High-Fidelity DNA Polymerase (Thermo Scientific, Waltham, MA USA, cat#F530S), using 2 dpf zebrafish embryos cDNA as template and zTGs2-specific primers, designed to insert restriction sites for KpnI and HindIII, respectively, at the 5′ and 3′ of the coding sequences (Table 2). The PCR products were gel purified with the GenElute™ Gel Extraction Kit (Sigma-Aldrich, Burlington, MA, USA, cat#NA1010), following the manufacturer’s instructions, and digested with KpnI and HindIII (New England BioLabs, Ipswich, MA, USA, cat#R3142 and cat#R3104) together with the recipient plasmid. The digested vector and DNA fragments were ligated with T4 DNA ligase (Thermo Scientific, cat#EL0011) and cloned into DH5α competent cells. Plasmid minipreps were obtained from overnight culture at 37 °C in 2xTY medium with GenElute™ Plasmid Miniprep Kit (Sigma-Aldrich, cat#PLN70), and transformed into BL21(DE3) competent cells.

### 4.7. Recombinant Protein Production and Purification

BL21(DE3) bacteria with the pET30a(+) plasmid vector bearing the zTG2 coding sequence were plated on a fresh 2xTY agar plate with 50 µg/mL Kanamycin (Kan) and left to grow overnight (O/N) at 30 °C. For each zTG2, a single colony was inoculated in 5 mL of 2xTY broth with 50 µg/mL Kan and 1% (*v*/*v*) glucose. After 5 h of growth at 37 °C in an orbital shaker, the bacterial cultures were diluted 1:100 in 2xTY with Kan 50 µg/mL and left growing at 25 °C in an orbital shaker. When the optical density at 600 nm (OD600nm) was about 0.6/0.8, the recombinant protein production was induced with 0.2 mM isopropyl-β-D-1-thiogalattopiranoside (IPTG) and was carried out for 16 h at 25 °C. The bacterial cultures were centrifuged at 4500× *g* for 15 min at 10 °C. Lysis of the bacterial pellet was performed in lysis buffer (20 mM Tris/HCl pH 8, 500 mM NaCl, 1% Triton X100, 5 mM imidazole, 100 µg/mL lysozyme (Sigma-Aldrich, cat#34046) and 50 µg/mL DNase I (Sigma-Aldrich, D5025), supplemented with cOmplete™ Protease Inhibitor Cocktail Tablets (Roche, Rotkreuz, Switzerland, cat#04693116001). The bacterial lysate was sonicated three times at 30% amplitude for 30″ and then centrifuged at 11,000× *g* for 30 min at 10 °C. Supernatant was recovered and the recombinant 6His-Tagged protein was column affinity-purified with HisPur™ Ni-NTA Resin (Thermo Scientific cat#88221). All purification steps were executed with a buffer composed of 20 mM Tris/HCl pH8, 500 mM NaCl. In the equilibration phase, 10 mM imidazole was added. In the wash step, 20 mM imidazole was used, and while in the elution step, 300 mM imidazole was used. The purified proteins were dialyzed for 16 h at 8 °C in Tris/Acetate-EDTA buffer supplemented with 5 mM dithiothreitol (DTT). 

### 4.8. zTGs2 Transamidation Activity Curve and Inhibition Assays

ELISA wells were coated O/N at 4 °C for 16 h with 120 µL of 15 µg/mL N,N′-dimethyl casein (DMC) in PBS. After one wash with PBS, 100 µL of reaction mix (100 mM Tris-HCl, pH 8.0, 0.2 mM EZ-Link™ Pentylamine-Biotin from Thermo Scientific cat#21345, 10 mM DTT, 5 mM CaCl_2_) were added according to the test: (A) with increasing amounts of recombinant TG2; (B) with 50 ng of recombinant TG2 and increasing concentrations of CaCl_2_; (C) with 50 ng of recombinant TG2 in the presence or in the absence of 100 µM guanosine 5′- triphosphate (GTP); (D) with 50 ng of recombinant TG2 in the presence or in the absence of hTG2-specific peptide inhibitors 1-155, R281 or ZDON (Sigma-Aldrich, cat#616467) at three different concentrations (25 mM, 2.5 mM and 0.25 mM). Samples were added in each well and the plate was incubated for 1 h at 37 °C. Following three washes with PBS-Tween 20 0.1% and three with PBS, the plate was incubated for 1 h at 37 °C with 100 µL per well of Pierce™ High Sensitivity Streptavidin-HRP (Thermo Scientific, cat#21134) diluted in a solution of 1% BSA in PBS in volumetric ratio 1:200. After washes, the colorimetric reaction was developed by adding 100 µL per well of TMB and stopped after 5 min by adding 50 µL per well of 2.5 mM H_2_SO_4_. The absorbance was measured at 450 nm using a microplate reader.

### 4.9. RGD-Independent Cell Adhesion Assay

Tissue culture plastic was coated overnight at 4 °C with 50 μL of human fibronectin (Sigma-Aldrich, cat# F0895) 5 μg/mL in 50 mM Tris-HCl, pH 7.4. After one wash with 100 μL of 50 mM Tris-HCl, pH 7.4, the fibronectin (FN) matrix was blocked with 100 μL of 3% (*w*/*v*) heat-inactivated BSA in PBS for 30 min at 37°. After one wash, hTG2 and zTGs2 were immobilized on FN matrix by incubating 100 μL of 5 μg/mL TGs2 in 2 mM EDTA in PBS, pH 7.4 at 37 °C for 1 h. Exponentially growing human foreskin dermal fibroblasts (HFDFs) were detached with 0.25% (*w*/*v*) trypsin in 5 mM EDTA, washed twice in serum-free medium to remove the traces of serum proteins and incubated with or without 150 μg/mL GRGDTP synthetic peptide for 20 min at 37 °C in a 5% CO_2_, 95% (*v*/*v*) air atmosphere. Then, cells were seeded on matrices at a density of 3 × 105 cells/mL for a maximum of 20 min to minimize the secretion of any endogenous protein at 37 °C in a 5% CO_2_, 95% (*v*/*v*) air atmosphere. After 1 gentle wash with PBS to remove non-attached cells, the cells were fixed in 3.7% (*w*/*v*) paraformaldehyde in PBS for 15 min at room temperature, then washed twice with PBS and permeabilized in 0.1% (*v*/*v*) Triton X-100 in PBS for 15 min at room temperature. After two washes with PBS, the cells were stained first with May–Grunwald stain, then with Giemsa stain. Digital images of 3 non-overlapping fields covering the central portion of each well were captured using a video digital camera (Nikon E5400). At least 9 images of separate fields per sample were examined.

### 4.10. Actin Cytoskeletal Staining

Nunc™ Lab-Tek™ Chamber Slide System (Thermo Scientific, cat#177445) was coated overnight at 4 °C with 150 μL of 5 μg/mL FN in 50 mM Tris-HCl, pH 7.4. After one wash with 200 μL of 50 mM Tris-HCl, pH 7.4, the FN matrix was blocked with 200 μL of PBS buffer supplemented with 3% (*w*/*v*) heat-inactivated BSA for 30 min at 37°. After one wash, hTG2 and zTGs2 were immobilized on FN matrix by incubating 200 μL of 5 μg/mL TGs2 in 2 mM EDTA in PBS, pH 7.4 at 37 °C for 1 h. Exponentially growing HFDFs were detached with 0.25% (*w*/*v*) trypsin in 5 mM EDTA, washed twice in serum-free medium to remove the traces of serum proteins and incubated with or without 150 μg/mL GRGDTP synthetic peptide for 20 min at 37 °C in a 5% CO_2_, 95% (*v*/*v*) air atmosphere. Then, cells were seeded on matrices at a density of 6 × 10^4^ cells per well for a maximum of 20 min to minimize the secretion of any endogenous protein at 37 °C in a 5% CO_2_, 95% (*v*/*v*) air atmosphere. After 1 gentle wash with PBS to remove non-attached cells, the cells were fixed in 3.7% (*w*/*v*) paraformaldehyde in PBS for 15 min at room temperature, then washed twice with PBS and permeabilized in 0.1% (*v*/*v*) Triton X-100 in PBS for 15 min at room temperature. Cells were then blocked in PBS buffer supplemented with 3% (*w*/*v*) heat-inactivated BSA and then incubated with FITC-labelled phalloidin (20 μg/mL) in blocking buffer. After three washes with PBS, coverslips were mounted with Vectashield mountant containing propidium iodide (Vector Laboratories) and examined by epifluorescence microscopy. 

### 4.11. Molecular Cloning of Zebrafish Transglutaminase 2 Sequence into Eukaryotic Expression Vector

The hTG2 and zTGs2 coding sequences have been cloned in a modified version of pCDNA 3.1 HYGRO(+) eukaryotic expression vector, in order to express the TGs2 as SV5-tagged proteins in HEK293 cells. Zebrafish TGs CDS were PCR-amplified with Phusion™ High-Fidelity DNA Polymerase (Thermo Scientific, cat#F530S), using pET30a(+) plasmid vector bearing the TGs2 coding sequences as template and TGs2-specific primers, designed to insert restriction sites for XbaI or NheI and HindIII, respectively, at the 5′ and 3′ of the coding sequences, as well as the nucleotide sequence encoding the SV5 tag (the primers’ sequences are reported in Table 3). The PCR products were gel purified with the GenElute™ Gel Extraction Kit, following the manufacturer’s instructions, and digested with XbaI or NheI and HindIII (New England BioLabs, cat#R0145, cat#R3131, and cat#R3104) together with the recipient plasmid. The digested vector and DNA fragments were ligated with T4 DNA ligase and cloned into DH5α competent cells. Plasmid minipreps were obtained from overnight culture at 37 °C in 2xTY medium with GenElute™ Plasmid Miniprep Kit.

### 4.12. Transient Cell Transfection and Selection of Stable TG2-Expressing HEK293 Cells

HEK293 cells were cultured in DMEM ((Thermo Scientific, cat#41966029) supplemented with 10% (*w*/*v*) FBS and 1% (*w*/*v*) Penicillin-Streptomycin ((Thermo Scientific, cat#15140122). For transfection, HEK293 cells were seeded in 6-well plate and eukaryotic expression vector pCDNA 3.1 HYGRO(+) containing the sequence encoding hTG2 or zTGs2 fused to SV5 tag was transfected with LipofectamineTM 2000 Transfection Reagent (Invitrogen, cat#11668019), according to the manufacturer’s instructions. After 48 h, the cells were either detached with 0.25% (*w*/*v*) trypsin in 5 mM EDTA for cell lysis or cultured with culture medium supplemented with 200 μg/mL Hygromycin B Gold (InvivoGen Toulouse, France).

### 4.13. HEK293 Anti-SV5 Tag Western Blot

HEK293 cells expressing hTG2, zTGs2, or the non-transfected negative control cells were lysed in lysis buffer (Tris-HCl 20 mM pH 8, NaCl 500 mM, 1% Triton X-100, supplemented with cOmplete™ Protease Inhibitor Cocktail Tablets), sonicated, and centrifuged at 14,000× *g* for 15 min at 4 °C. The clear protein lysates were collected, quantified with Bradford reagent (Sigma-Aldrich, cat#B6916), the proteins were resolved on SDS-polyacrylamide gel and transferred onto a nitrocellulose membrane. After a blocking step of 45 min with 5% skim milk in PBS at room temperature (RT), the membrane was incubated for 1 h with an anti-SV5 antibody. After three washes with PBS-Tween 20 0.1% and three with PBS, the membrane was incubated with anti-mouse IgG antibody (Jackson ImmunoResearch, Ely, UK) diluted in the blocking buffer in volumetric ratio 1:2500. After three washes with PBS-Tween 20 0.1% and three with PBS, the membrane was developed with Pierce™ ECL Western Blotting Substrate (Thermo Scientific, cat#32209) and imaged with ChemiDoc™ MP Imaging System (Bio-Rad).

### 4.14. HEK293 Cell Lysates Transamidation Activity Assay

ELISA wells were coated O/N at 4 °C for 16 h with 120 µL of 15 µg/mL N,N′-dimethyl casein (DMC) in PBS. After one wash with PBS, 100 µL of reaction mix (100 mM Tris-HCl, pH 8.0, 0.2 mM EZ-Link™ Pentylamine-Biotin from Thermo Scientific cat#21345, 10 mM DTT, 5 mM CaCl_2_) with HEK293 cell protein lysates were added in each well and the plate was incubated for 1 h at 37 °C. For protein lysates of HEK293 cells after transient transfection, 1 μg of lysate per well was used. For protein lysates of hygromycin-selected HEK293 stable clones expressing TGs2, the quantity of samples used was normalized on the level of TG2 expression, measured by quantifying the relative protein band on the nitrocellulose membrane with Image Lab Software version 6.1 (Bio-Rad). Following three washes with PBS-Tween 20 0.1% and three with PBS, the plate was incubated for 1 h at 37 °C with 100 µL per well of Pierce™ High Sensitivity Streptavidin-HRP diluted in a solution of 1% BSA in PBS in volumetric ratio 1:200. After washes, the colorimetric reaction was developed by adding 100 µL per well of TMB and stopped after 5 min by adding 50 µL per well of 2.5 mM H_2_SO_4_. The absorbance was measured at 450.

### 4.15. HEK293 Cells Flow Cytometry Analysis

HEK293 cells expressing TGs2 and wild-type HEK293 were detached with 0.25% (*w*/*v*) trypsin in 5 mM EDTA, washed in PBS, and fixed in 3.7% (*w*/*v*) paraformaldehyde in PBS for 15 min at room temperature, then washed twice with PBS and permeabilized in PBS supplemented with 0.1% (*v*/*v*) Triton X-100 and 2% (*w*/*v*) BSA for 30 min at room temperature. After two washes, cells were stained with anti-SV5 tag antibody conjugated with phycoerythrin (α-SV5-PE) on ice for 30 min, washed and analysed with Attune NxT Flow Cytometer (Invitrogen).

### 4.16. Apoptosis Assays with Hydrogen Peroxide and Thapsigargin

Preliminary experiments were performed with different time points and concentrations of inducer to determine the best conditions for induction of apoptosis. 

For H_2_0_2_ treatment, HEK293 cells expressing TGs2 and wild-type HEK293 cells were seeded in a 24-well plate in a culture medium. The next day, the cell medium was replaced with a new one supplemented with 2% FBS, and after 16 h, cells were treated adding 0.4 mM H_2_0_2_ for 2 h. Untreated cells were used as reference control. 

For Thapsigargin treatment, HEK293 cells were seeded in a 24-well plate in a culture medium, with 10% FBS. The next day, cells were washed once with serum-free medium and then treated with 5 μM thapsigargin (Sigma-Aldrich, cat#T9033) in serum-free medium for 24 h. Untreated cells were used as reference control.

After each treatment, the cells were detached with 0.25% (*w*/*v*) trypsin in 5 mM EDTA, washed in annexin binding buffer (HEPES 20 mM, 150 mM NaCl, 2.5 CaCl_2_), and stained for 15 min with Annexin V-fluorescein isothiocyanate (FITC) (ImmunoTools, Friesoythe; Germany, cat#31490013) and Propidium Iodide (PI) Ready-Flow^TM^ Reagent (Invitrogen^TM^, cat#R37169). Unstained wild-type cells and wild-type cells stained with Annexin V-FITC only and PI only were used as control samples for fluorescence compensation. Cells were analysed with Attune NxT Flow Cytometer (Invitrogen).

### 4.17. Caspase3 Activity Assay

Caspase activity was evaluated at the end of the apoptosis assay performed with H_2_O_2_ or Thapsigargin. In detail, after treatment, the cells were detached with 0.25% (*w*/*v*) trypsin in 5 mM EDTA, washed with PBS, and incubated in a 1:1 ratio (*v*/*v*) with Caspase 3/7 Glo^®^ (Promega, Madison, WI, USA) in a 96-well black plate (PerkinElmer, Milano, Italy) for 1 h at RT in the dark. Finally, the luminescence signal was detected with Promega/Glomax multi-detection system.

## Figures and Tables

**Figure 1 ijms-24-12041-f001:**
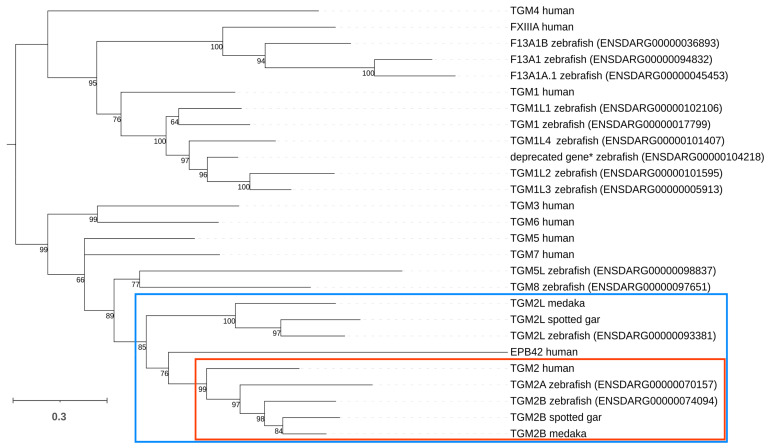
Zebrafish transglutaminase gene family. Midpoint-rooted maximum likelihood phylogeny of zebrafish *tgm* family genes, built based on the multiple sequence alignment of deduced amino acid sequences. To allow a reliable interpretation of orthology/paralogy relationships among sequences, all human *TGM* and a few selected fish *tgm2* and *tgm2l* sequences were added. Bootstrap support values are shown close to each node. Nodes supported by bootstrap values < 50 were collapsed. The blue and red boxes highlight the monophyletic clades including all *TGM2* and *tgm2l* sequences, and *TGM2* orthologs only, respectively. * ENSDARG00000104218 is not included in the most recent Ensembl gene annotation release.

**Figure 2 ijms-24-12041-f002:**
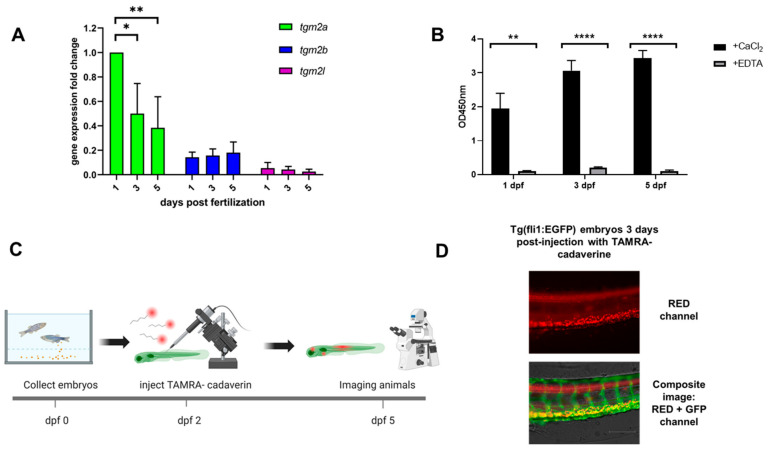
The three TG2 genes of zebrafish are expressed during embryonic development. (**A**) RT-qPCR expression analysis of zebrafish transglutaminase 2 genes. The graph shows the change in gene expression of the three zebrafish TG2 orthologs at 1, 3, and 5 dpf. Expression data are normalized against the zebrafish β-actin gene. * = *p* < 0.05, ** = *p* < 0.01. **** = *p* < 0.0001. (**B**) Transamidation activity of soluble protein lysates from zebrafish embryos was assayed by ELISA with and without 20 mM EDTA at 1, 3, and 5 dpf. (**C**) Schematic representation of the experimental protocols. Two dpf Tg(fli1:EGFP) embryos were microinjected into the yolk sac *with TAMRA-cadaverine. At 3 days after injection, the incorporated cadaverine was observed by epifluorescence microscopy. Created with BioRender.com. (**D**) Representative images show the trunk region in the red channel showing cadaverine incorporation and a composite image of the red and green channels showing the embryo vasculature.

**Figure 3 ijms-24-12041-f003:**
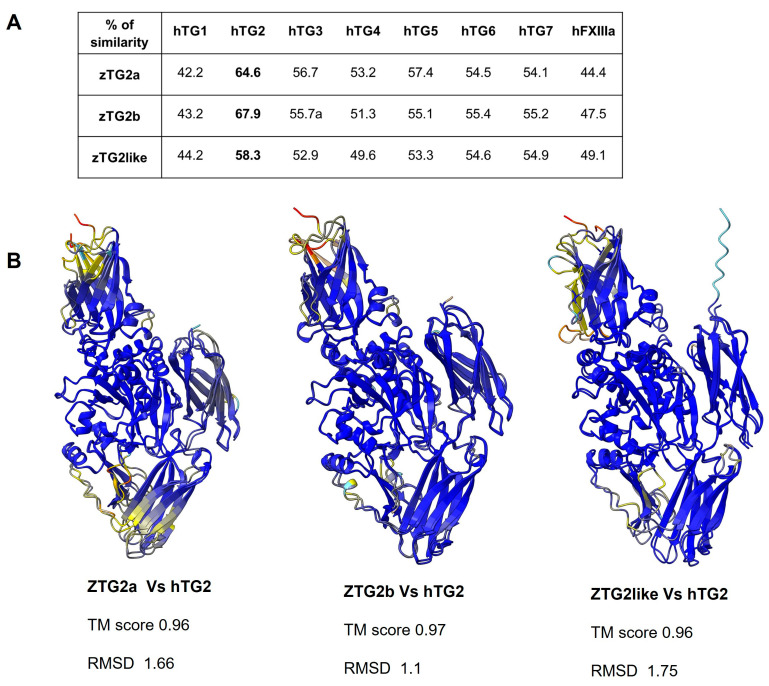
Sequence and structure comparison between zebrafish and human TG2. (**A**) Table shows the percent similarity between the three zebrafish TG2 orthologues and the human TGs, calculated using pairwise sequence. (**B**) 3D models showing the superpositions of the three zebrafish TGs2 structures with the human TG2. The 3D models were predicted using the AlphaFold tool and rendered using ChimeraX 1.6.1 software. The models show the degree of structural divergence with a color scale representing RMSD values from blue (highly similar, low RMSD) to yellow (somewhat similar) to red (highly divergent). The overall RMSD and TM-score of the three aligned pairs are also given under each model.

**Figure 4 ijms-24-12041-f004:**
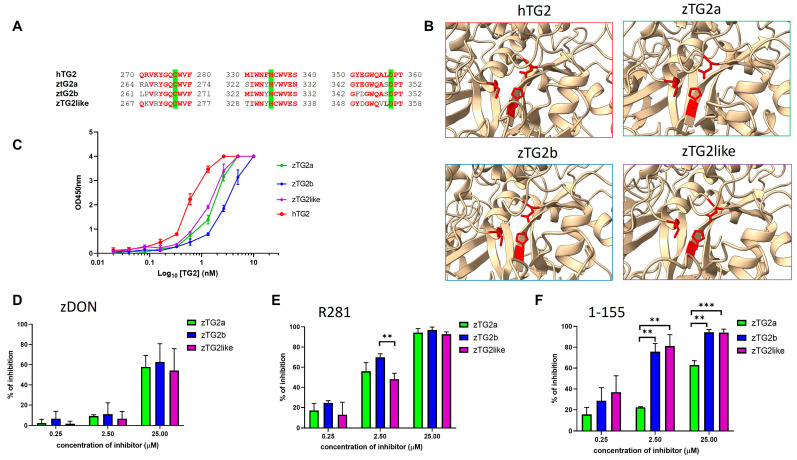
Enzymatic activity of TG2 (**A**) Multiple protein sequence alignment of the zTGs2 proteins and the hTG2. The catalytic triad Cys-His-Asp, highlighted in green, is conserved as many or the nearby residues that are in bold and highlighted in red. (**B**) 3D models of the human and zebrafish TGs2 active site, rendered with ChimeraX. The catalytic triad Cys-His-Asp is colored in red. (**C**) The three affinity-purified zTGs2 recombinant proteins were tested at increasing concentrations of proteins in ELISA by measuring the TG2-catalysed incorporation of pentylamine-biotin into DMC. Recombinant hTG2 was also used as positive control. The 450 nm absorbance values are plotted against the Log_10_ of the TGs2 concentration. The inhibitors were tested at 0.25, 2.5 and 25 μM. (**D**) ZDON (Zedira); (**E**) R281 and (**F**) 1-155. The graphs represent percentage of inhibition. Each plot represents the averaged result of three independent experiments (*n* = 3). ** = *p* < 0.01. *** = *p* < 0.001.

**Figure 5 ijms-24-12041-f005:**
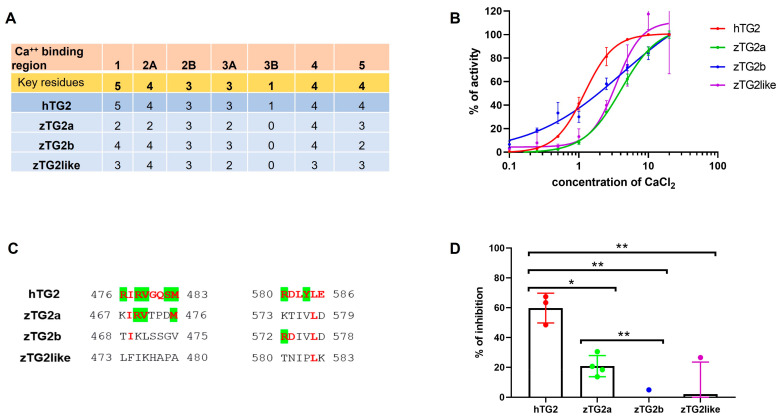
Calcium and GTP-dependent activity assays. (**A**). The table includes in the first two rows the names of the 5 putative Ca^2+^ binding sites identified by Kiraly and colleagues [37] and the number of key residues at each site. Each row below shows the number of conserved key residues at the same positions for hTG2 and the three zTG2 (**B**) The TGs2-catalysed incorporation of pentylamine-biotin into coated DMC was tested at increasing concentrations of CaCl_2_. The plot represents the data as percentage of activity with respect to the sample tested with the highest concentration of calcium. (**C**) Multiple protein sequence alignment of the putative GDP/GTP-binding site of hTG2 and zTGs2. The key residues identified as the responsible for the interaction are highlighted in green and the conserved residues in bold red. (**D**) GTP-dependent enzymatic inhibition assay. The transamidation activity of hTG2 and zTGs2 have been tested in the presence and absence of 100 mM GTP. The decrease in TG activity is represented as percentage of enzymatic inhibition. The plot represents the averaged result of three independent experiments (*n* = 3). * = *p* < 0.05; ** = *p* < 0.01.

**Figure 6 ijms-24-12041-f006:**
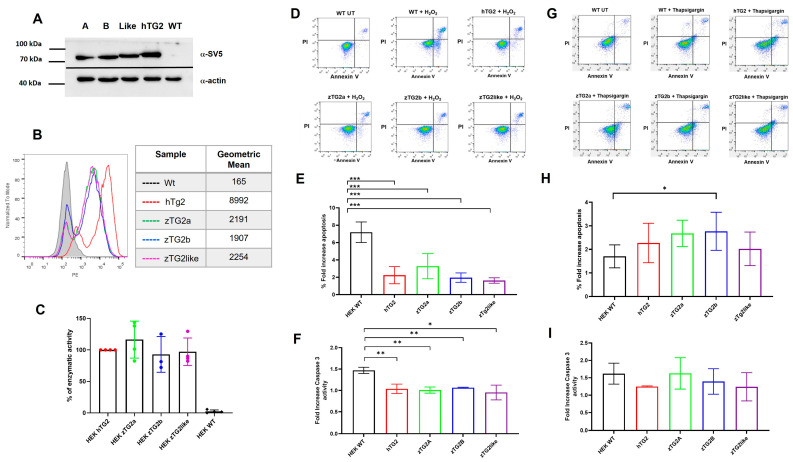
Zebrafish TGs2 activity in HEK293T cell model. HEK293 cells with overexpression of zTGs2 and hTG2 were isolated by transfection and selection with hygromycin. The expression of SV5-tagged protein was confirmed with α-SV5 tag antibody in (**A**) Western Blot on cell lysate and (**B**) flow cytometry detecting intracellular TG2 with V5 antibody on fixed and permeabilized stable cell clones. (**C**) Enzymatic activity was measured using cell lysate and normalized on the expression level of the TGs2. The data are expressed as a percentage of enzymatic activity relative to the hTG2-expressing HEK293 cell lysate, used as positive control. Apoptosis was induced in HEK293 cells expressing zTGs2, hTG2 and wild-type HEK293 by treatment with 0.4 mM H_2_O_2_ for 2 h (**D**–**F**) or 5 μM thapsigargin for 24 h (**G**–**I**). Treated cells were stained with Annexin V-FITC and propidium iodide (PI) and analyzed by flow cytometry to assess apoptosis. Dot plot representative experiment Annexin V and PI staining of treated samples ((**D**) for H_2_O_2_ and (**G**) for thapsigargin) and histogram plot of the percentage of Annexin V-FITC positive cells ((**E**) and (**H**) panel). Caspase 3 activity was measured directly on treated cells using Caspase 3/7 Glo^®^ (Promega) ((**F**) for H_2_O_2_ and (**I**) for thapsigargin). * = *p* < 0.05, ** = *p* < 0.01. *** = *p* < 0.001.

**Figure 7 ijms-24-12041-f007:**
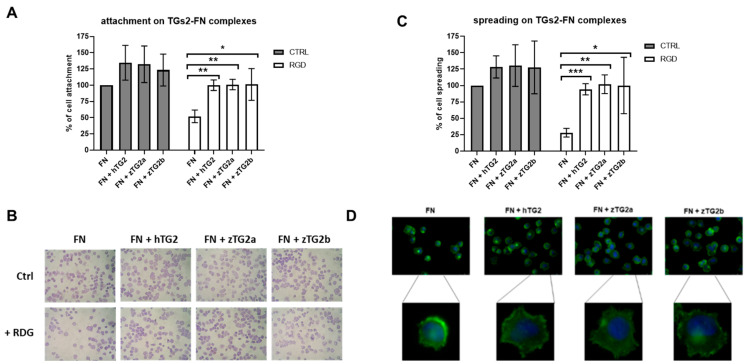
Zebrafish TGs2 mediate RGD-independent cell adhesion of HFDFs. Cell attachment (**A**) and cell spreading (**B**) on FN and TGs2-FN complexes were assessed ~20 min after seeding cells pre-incubated with 150 μg/mL GRGDTP synthetic peptide (RGD) (white bars) or serum-free medium (CTRL, grey bars). The plots represent the averaged results of three independent experiments. Data are expressed as mean ± S.D. of the percentage of cell attachment or cell spreading, compared to the control values on FN (100%). Mean attached cells value on FN control was 96 ± 25; mean spread cells value on FN control was 52 ± 10. The average number of total cells analyzed for the control sample was 543. *= *p* < 0.05; ** = *p* < 0.01; *** = *p* < 0.001. (**C**) microscopy of HFDFs seeded on FN and FN-TGs2 complexes after incubation with RGD peptide or control. Representative images of HFDFs seeded on FN and FN-TGs2 complexes after incubation with RGD peptide or control. Images after fixation, permeabilization, and May–Grunwald and Giemsa staining for the cytoplasm and for the nucleus, respectively (100× magnification). (**D**) Epifluorescence microscopy of HFDFs seeded on FN and FN-TGs2 complexes, after incubation with RGD peptide or serum-free medium. Actin stress fibers were visualized using fluorescein isothiocyanate (FITC)-labelled phalloidin (200× magnification).

**Table 1 ijms-24-12041-t001:** List of primers used for the RT-qPCR experiments.

Primer Name	Sequence 5′-3′
zTG2a Forward	CCAAAGCAGTGGGTCGAGAT
zTG2a Reverse	GTGCGTAAAACATCACCCGG
zTG2b Forward	GATGGGTCGTTCCGATCTCC
zTG2b Reverse	GCTGATCTTCTGGCCCACTT
zTG2like Forward	CACCTGCAGCCCTGAAAAAC
zTG2like Reverse	TCTGGTTGGGATGCCAAGAC
Beta actin Forward	CGAGCTGTCTTCCCATCCA
Beta actin Reverse	TCACCAACGTAGCTGTCTTTCTG
Lsm12b Forward	AGTTGTCCCAAGCCTATGCAATCAG
Lsm12b Reverse	CCACTCAGGAGGATAAAGACGAGTC
Hsp70 Forward	TCAAGCGCAACACAACCATC
Hsp70 Reverse	ATTTGCCCAGCAGGTTGTTG

**Table 2 ijms-24-12041-t002:** List of primers used for the cloning of the three zTG2 coding sequences.

Primer Name	Sequnce 5′-3′
zTG2a sense KpnI	AGATCTGGGTACCATGGAGAGAGTGGTGGAG
zTG2a anti HindIII	TCCTCGAGAAGCTTTTATTCATAGATAACCAG
zTG2b sense KpnI	AGATCTGGGTACCATGGCTCTGGACATCGGC
zTG2b anti HindIII	TCCTCGAGAAGCTTTCATTTCCCGATGATGAC
zTG2like sense KpnI	AGATCTGGGTACCATGGCCAGCTATAATGCC
zTG2like anti HindIII	TCCTCGAGAAGCTTCTAAATTTCTGGGACAGC

**Table 3 ijms-24-12041-t003:** List of primers used for the cloning the TG2 coding sequences into eukaryotic vector.

Primer Name	Sequence 5′-3′
hTG2-HYGRO-Sense_NheI	GCTGGCTAGCTGCCACCATGGCCGAGGAGCTGGTC
hTG2-HYGRO-anti_SV5	GATTGGTTTGCCACTAGTGGCGGGGCCAATGATGAC
zTG2a-HYGRO-Sense_Xba	AGCTGTCTAGATGCCACCATGGAGAGAGTGGTGG
zTG2a-HYGRO-anti_SV5	GATTGGTTTGCCACTAGTTTCATAGATAACCAGATTC
zTG2b-HYGRO-Sense_NheI	GCTGGCTAGCTGCCACCATGGCTCTGGACATCGGC
zTG2b-HYGRO-anti_SV5	GATTGGTTTGCCACTAGTTTTCCCGATGATGACGTTC
zTG2like-HYGRO-Sense_Xba	GCTGTCTAGATGCCACCATGGCCAGCTATAATGCC
zTG2like-HYGRO-anti_SV5	GATTGGTTTGCCACTAGTAATTTCTGGGACAGCATTTAC
SV5_HindIII_anti	AGCTAAGCTTTTAAGTACTATCCAGGCCCAGCAG TGGGTTTGGGATTGGTTTGCCACTAGT

## Data Availability

Data supporting reported results are available from the coordinator of the study, Prof. Daniele Sblattero, corresponding author.

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
