# Peer review of "Biochemical and Functional Characterization of the Three Zebrafish Transglutaminases 2"

_ijms, 2023, doi:10.3390/ijms241512041_

Round 1

Reviewer 1 Report

Dear Authors,

Congratulations for your effort in putting up a good experimental paper. The experiments were clean and well performed.

However, I have some inputs if those can be incorporated. 

1. Fig 2 a. requires stats.

2. Fig 4. requires stats. All the bar graphs.

3. Can Figure 6B, on the Y axis have modal representation. With the geometric means of the channels.

4. Line 671 H2SO4 sub script issue.

5. in material and method section "Apoptosis assays with hydrogen peroxide and Thapsigargin"..please make it clear why the time points are different. The paragraph needs a bit rephrasing about H2O2 treatment and thapsigargin.

6. Promega/Glomax multi-detection system? Was this read in a tecan?

7. Briefly mention the role of these  ZDON etc inhibitors in the discussion.

8. In the alphafold generated model can you please highlight the important residue with aa no. instead of just highlighting it.

10. Typographical errors here and there.

Author Response

1- Fig 2 a. requires stats.

DONE. This figure has been updated including Statistics.

2- Fig 4. requires stats. All the bar graphs.

DONE. This figure has been updated including Statistics.

3- Can Figure 6B, on the Y axis have modal representation. With the geometric means of the channels.

DONE. We have updated the figure according to the suggestion

4- Line 671 H2SO4 sub script issue.

DONE. Corrected

5- in material and method section "Apoptosis assays with hydrogen peroxide and Thapsigargin"..please make it clear why the time points are different. The paragraph needs a bit rephrasing about H2O2 treatment and thapsigargin.

DONE, we have corrected the error for time exposure and rewritten the paragraph to improve clarity.

6- Promega/Glomax multi-detection system? Was this read in a tecan?

The Promega/Glomax multi-detection system is a multipurpouse plate reader. In this case we have use it as luminometer to detect and record the signal using internal standard program.

7- Briefly mention the role of these ZDON etc inhibitors in the discussion.

Done, We have added some sentences on the role of TG2 inhibitors in the discussion

8-In the alphafold generated model can you please highlight the important residue with aa no. instead of just highlighting it.

Thanks for the suggestion. We tried to create the proposed figure, but since there are at least 10 regions in each protein alignment that have significant RMSD, the figure unfortunately became quite unreadable. We believe that our version with the RMDS color code is still the best way to show the structural differences. If requested we can prepare a version with AA number. 

9- Typographical errors here and there.

We revised again the text and correct as typos and errors.

Reviewer 2 Report

The manuscript characterized the enzymatic and functional properties of TG2 proteins expressed in zebrafish. and the results indicated that TGs2   proteins behave very similarly to the human ortholog and pave the way for future in vivo studies of 28 TG2 functions in zebrafish. The manuscript is well written while some minor issues are listed as follow.

1. The same background should be presented in Figure 4B, as well as the distances between the key atoms of the active triad.

2. As the manuscript inferred, Figure 5A was not the author's result and should not presented in the Results Section.

3. Some formats error were found in Line 716-717

4. The 3D structures of the cloned proteins were modeled using Alphafold, while not dissussed and evaluated in detail.

Author Response

  1. The same background should be presented in Figure 4B, as well as the distances between the key atoms of the active triad.

DONE figure have been redone.

  1. As the manuscript inferred, Figure 5A was not the author's result and should not presented in the Results Section.

DONE, Thanks for the suggestion. We have substituted the alignment figure with a table summarizing the number of Key residues present in each zebrafish, which could be more informative.

  1. Some formats error were found in Line 716-717

Done. The errors are corrected, and text is simplified.

  1. The 3D structures of the cloned proteins were modeled using Alphafold, while not dissussed and evaluated in detail.

OK we have added some phrases on the models.